# Telehealth Utilization and Good Care among Informal Caregivers: Health Information National Trends Survey, 2022

**DOI:** 10.3390/healthcare11243193

**Published:** 2023-12-18

**Authors:** Zahra Mojtahedi, Ivan Sun, Jay J. Shen

**Affiliations:** 1Department of Healthcare Administration and Policy, School of Public Health, University of Nevada, Las Vegas, NV 89154, USA; mojtahed@unlv.nevada.edu (Z.M.); ivan.sun@unlv.edu (I.S.); 2UNLV Brookings Mountain West, University of Nevada, Las Vegas, NV 89154, USA; 3Center for Health Disparities and Research, School of Public Health, University of Nevada, Las Vegas, NV 89119, USA

**Keywords:** disparity, informal caregivers, HINTS, quality, race, surveys, telehealth

## Abstract

Background: The COVID-19 pandemic accelerated the adoption of telehealth services. Informal caregivers provide vital support to family and friends. Studying telehealth among informal caregivers is crucial to understanding how technology can support and enhance their caregiving responsibilities, potentially enhancing telehealth services for them as well as their patients. The present study aims to nationally investigate telehealth utilization and quality among informal caregivers. Methods: This cross-sectional investigation employed the 2022 Health Information National Trends Survey (HINTS) dataset. Informal caregivers, telehealth variables (utilization, good care, technical problems, convenience, and concerns about infection exposure), and sociodemographic factors (age, gender, race/ethnicity, income, education, health insurance, and census regions) were identified based on questions in the survey. Weighted multivariable logistic regression models were employed to calculate odds ratios (ORs), 95% confidence intervals (CIs), and *p*-values. Results: Significant disparities in telehealth utilization were detected among informal caregivers (N = 831), when telehealth users were compared to non-users. Those aged 50–64 (OR = 0.36, 95% CI = 0.20–0.65) and 65+ (OR = 0.40, 95% CI = 0.21–0.74) had significantly lower odds of using telehealth than those aged 35–49. Men had significantly lower odds of telehealth utilization (OR = 0.47, 95% CI = 0.25–0.87). Black caregivers compared to Whites had significantly lower odds (OR = 0.49, 95% CI = 0.24–0.99), while health insurance increased odds (OR = 5.31, 95% CI = 1.67–16.86) of telehealth utilization. Informal caregivers who used telehealth were more likely to be perceived as good telehealth caregivers if they had no telehealth technical issues compared to caregivers who had (OR = 4.61, CI = 1.61–13.16; *p*-value = 0.0051) and if they were from the South compared to the West (OR = 2.95, CI = 1.18–7.37, *p*-value = 0.0213). Conclusions: For the first time, to the best of our knowledge, we have nationally investigated telehealth utilization and quality among informal caregivers. Disparities in telehealth utilization among informal caregivers are evident, with age, gender, race, and health insurance being significant determinants. Telehealth quality is significantly influenced by technical problems and census regions, emphasizing the importance of addressing these aspects in telehealth service development for informal caregivers.

## 1. Introduction

Telehealth, the provision of healthcare services through virtual technology, has been significantly promoted since the COVID-19 pandemic [1], but its use and quality can be influenced by sociodemographics and other factors [2]. Telehealth allows patients and caregivers to access medical care, consult with healthcare professionals, and receive timely guidance while minimizing the risk of infection and eliminating transportation and childcare challenges [2]. Informal caregivers play an important role in ensuring the well-being of those they care for, usually family members and friends [3,4]. The use of telehealth by caregivers has been connected to a higher frequency of communication between caregivers and clinicians, as well as improved patient outcomes [5,6]. Identifying factors connected to telehealth use and quality among informal caregivers in a large sample size would be critical for developing policies that ensure equitable access to telehealth services, ultimately leading to improved disease outcomes.

Sociodemographic characteristics can critically influence the utilization of telehealth services, which are mostly studied in the overall patient population [7,8,9,10]. Women frequently use telehealth more than men, owing to their roles as primary caregivers and their increased familiarity with technology [7]. Age influences telehealth utilization, since older individuals may face more technological challenges, limiting their telehealth utilization [7]. The association of race/ethnicity and income with telehealth utilization has yielded mixed results [7,8,9,10,11,12]. From a theoretical point of view, individuals with higher income tend to have more resources, which facilitates telehealth consultations. These individuals are also more likely to have health insurance. Furthermore, people with higher education are more comfortable with technology, making them more inclined to use telehealth services [7]. Telehealth use was also found to be different based on census region, being highest among people residing in the Northeast and West regions [7]. More research is needed to identify disparities in telehealth utilization related to sociodemographic factors among informal caregivers to ensure equitable access to healthcare services for both caregivers and their patients.

Telehealth patient satisfaction and quality are less statistically and nationally studied than telehealth care utilization. In total, patient satisfaction with telehealth visits was reported to be high [3,13,14,15] and positively associated with certain sociodemographic factors, including female gender, younger age, non-White race/ethnicity, and lower education levels, while patient income showed mixed trends [16]. Patient satisfaction with telehealth has also been linked to comfort with technology, the convenience it offers, and the ability to avoid potential infection exposure [16,17]. Telehealth technical problems could disrupt the smooth flow of communication between parties, resulting in delayed or interrupted virtual sessions, affecting the quality of telehealth and patient satisfaction [16,17]. The convenience and reduced travel distances can contribute to the overall quality of healthcare delivery by promoting more frequent and timely interactions between caregivers and healthcare providers [16,17]. The avoidance of crowded waiting rooms and unnecessary exposure to potential sources of infection can also positively influence telehealth patient satisfaction [18]. However, most of the available telehealth-quality-related studies stem from small-scale, localized studies that focus on the overall patient population, which might be related to the absence of telehealth and/or caregiver-related questions in earlier national surveys. These questions have been recently added to national surveys, notably the Health Information National Trends Survey (HINTS). The survey was expanded to include questions on caregiving and telehealth in 2017 and 2022, respectively [19,20,21,22,23], providing a valuable quantitative resource for understanding the use and quality of telehealth among caregivers across the nation.

This study aimed to explore telehealth utilization and quality among informal caregivers at the national level by analyzing the HINTS dataset in 2022. In terms of telehealth utilization, it investigated disparities in sociodemographic characteristics, including age, gender, race/ethnicity, income, education, health insurance, and census region. Moreover, it assessed the association of perceived good telehealth care with sociodemographic characteristics, telehealth technical problems, telehealth convenience/distance, and concerns with exposure to infection. The findings provided valuable insights for healthcare policymakers to improve the utilization and quality of telehealth services for this valuable source of healthcare workforce.

## 2. Methods

### 2.1. Survey

A cross-sectional study was conducted using the publicly available HINTS dataset. The University of Nevada Las Vegas deemed this secondary data analysis of deidentified, publicly available data exempt from review. Launched in 2003, HINTS is a nationally recognized survey conducted periodically by the National Cancer Institute (NCI). Its target population is civilian, non-institutionalized adults aged ≥ 18 residing in the US [24]. The HINTS 6 (7 March 2022 to 8 November 2022) contains 475 questions in 17 sections, including section A (looking for health information), section B (using the internet to find information), section C (your health care), section D (telehealth), section E (medical records), section F (caregiving), section G (genetic testing), section H (your overall health), section J (environment and health), section K (social determinants of health), section L (health and nutrition), section M (physical activity and exercise), section N (tobacco products), sections O, P, and Q (cancer-related questions), and section R (you and your household) [24]. The HINTS 6 employed a rigorous methodology for sample selection, data collection, data management, and weighting and variance estimation [23,24].

The HINTS 6 survey employed a two-stage sampling strategy for sample selection. A stratified sample of addresses was initially chosen from a residential address file. In the subsequent stage, one adult was selected within each sampled household. Unlike previous HINTS samples, HINTS 6 featured an expanded sample design that included four sampling strata by further categorizing the traditional high- and low-minority strata based on rural and urban geographic areas. While the high-minority strata were still sampled at a higher rate, the rural strata among the low-minority strata were also sampled at a higher rate than the urban strata, ensuring an adequate representation of rural addresses in the overall sample [23,24].

Selected individuals were categorized into control and treatment groups. In the control group, respondents were given the choice to respond via paper or online, whereas the treatment group initially had online-only options, later incorporating paper surveys in subsequent mailings. The survey was available in both English and Spanish, and all participants, irrespective of the response mode, received a USD 2 pre-paid monetary incentive to boost participation. Additional incentives were provided to the control group for completing the survey online. Mail surveys offer a cost-effective means to reach a diverse population, provide flexibility for respondents to complete the survey at their convenience, and accommodate those without telephone access or those who prefer a private setting, leading to enhanced response rates and a representative study sample. The choice of online surveys was added in 2019 to examine the feasibility of encouraging a sufficient number of participants to transition to online data collection while sustaining, or even enhancing, response rates. The mailing procedures are detailed in the HINTS 6 methodological report [23,24].

For data management, each retuned paper questionnaire underwent scanning, verification, cleaning, and editing procedures. Simultaneously, each web questionnaire underwent assessment using the Survey Builder verification and cleaning processes, and the data were extracted into a unified database, subsequently edited alongside the paper data. Additionally, imputation procedures were applied to both web and paper data [24]. The final HINTS 6 sample consists of 6252 respondents; of them, 67 were considered partial completers who did not answer the entire survey (responded to 50–79% of questions) [24]. A questionnaire was considered to be complete if at least 80% of questions were answered. The overall household response rate was 28.1%. Further details on sampling are publicly available in the HINTS 6 Methodology Report [24].

Each surveyed adult was assigned a full-sample weight along with a set of 50 replicate weights. Replicate weights were generated using the ‘delete one’ jackknife (JK1) replication method [24]. The full-sample weight and replicate weights were utilized for estimating population/subpopulation estimates and standard errors for these estimates, respectively. The application of sampling weights ensured accurate inferences from the respondent sample to the broader population, addressing nonresponse and noncoverage biases as much as possible. Calibration of person-level weights to population counts (referred to as control totals) was performed using estimates from the United States Census Bureau’s 2021 American Community Survey with the following variables: age, gender, education, marital status, race, ethnicity, and census region [24]. The 2021 National Health Interview Survey was also used to calibrate HINTS 6 data control totals regarding percent with health insurance [24].

### 2.2. Definition of Variables

#### 2.2.1. Informal Caregiving

Caregiving status was identified by this question: “Are you currently caring for or making health care decisions for someone with a medical, behavioral, disability, or other condition?” [23]. Those who responded “yes” were defined as caregivers. Professional caregivers were defined as those who responded “yes” to the following question: “(For the individual to whom you provide the most care) Do you provide any of this care professionally as a part of a job (for example, as a nurse or professional home health aide)?”. Among caregivers, professional caregivers were excluded. The remaining caregivers were defined as informal caregivers.

#### 2.2.2. Telehealth Utilization

A telehealth recipient was defined by this question: “In the past 12 months, did you receive care from a doctor or health professional using telehealth?” [23]. Those who responded “yes, by video”; “yes, by phone call (voice only with no video)”; or “Yes, some by video and some by phone call” were defined as telehealth recipients. It is worth noting that telehealth includes a variety of digital communications, notably through video and/or audio communications [2].

#### 2.2.3. Perceived Good Telehealth Care

Perceived good telehealth care was defined by this question: “Regarding your telehealth visits, how much do you agree or disagree—The care I received from telehealth was as good as a regular in-person visit”. Those who responded “strongly agree” were categorized as the good telehealth care group. Other caregivers who responded “somewhat agree”, “somewhat disagree”, or “strongly disagree” were categorized as the group with challenges in good telehealth care [22].

#### 2.2.4. Telehealth Technical Problems

Telehealth technical problems were defined by this question: “Regarding your telehealth visits, how much do you agree or disagree—I had technical problems with my telehealth visit(s) (for example, difficulty using the technology, trouble seeing or hearing my healthcare provider)” [22]. Those who responded “strongly disagree” were categorized as users with no telehealth technical problems. Other caregivers who responded “somewhat disagree”, “somewhat agree”, or “strongly agreed” were categorized as users with telehealth technical problems.

#### 2.2.5. Infection Exposure

A telehealth user whose reason for a telehealth visit was limiting infection exposure was defined by this question: “Why did you choose a telehealth visit(s)…I wanted to avoid possible infection at the doctor office or hospital (for example, COVID-19 or flu)” [22]. Those who responded “yes” were categorized as infection-aware telehealth users. Caregivers who responded “no” were categorized as infection-unaware telehealth users.

#### 2.2.6. Telehealth Convenience/Distance

Telehealth convenience was defined by this question: “Why did you choose a telehealth visit. It was more convenient than going to the doctor (for example, less travel or wait times)” [22]. Those who responded “yes” were categorized as convenience-oriented telehealth users. Caregivers who responded “no” were categorized as non-convenience-oriented telehealth users.

#### 2.2.7. Sociodemographic Characteristics

The sociodemographic characteristics were categorized based on available groups in the HINTS 6 codebook and also available literature from the HINTS dataset on caregivers [19,23]. They are indicated in Table 1 and include age, gender, race/ethnicity, income, education, health insurance, and census region.

### 2.3. Statistical Analyses

Data analyses were conducted using SAS OnDemand for Academics. SAS codes were based on the guidelines provided in the “overview of the HINTS 6 survey (2022) and data analysis recommendations” handbook [23]. Missing values were replaced with dots (.) using SAS codes [23]. Descriptive analyses were performed to summarize the frequency of each variable.

The two outcomes were telehealth utilization (+/−) and perceived good telehealth care (+/−). For both outcomes, weighted multivariable logistic regression models (proc surveylogistic) were used to calculate odds ratios (ORs) and 95% confidence intervals (CIs) [24]. Predictors for the telehealth utilization model were sociodemographic characteristics, including age, gender, race/ethnicity, income, education, health insurance, and census region. For the perception of good telehealth care, besides the sociodemographic characteristics, telehealth technical problems, telehealth convenience, and infection exposure were predictor variables. Jackknife replications were included in both models for variance estimation. The degrees of freedom for statistical testing were 49, as recommended by the data analysis handbook [24]. *p*-values less than 0.05 were considered significant in all analyses.

## 3. Results

There were 937 caregivers; of them, 106 were professional caregivers. Professional caregivers were excluded from the study since their differences from informal caregivers have been documented [25]. In total, 831 informal caregivers were available in the dataset. Of them, 13 had missing values for telehealth.

Table 1 depicts the descriptive characteristics of informal caregivers by telehealth utilization in the USA using HINTS 6, 2022. Although caregivers between the ages of 50 and 64 made up the largest group (34.0%) of telehealth users, closely followed by those aged 65 and older (31.0%), these age groups had lower parentages when compared to non-telehealth users. Compared to the percentage of non-telehealth users, telehealth users had a lower percentage for men, Black race/ethnicity, income levels less than USD 20,000, income levels between USD 50,000 and USD 75,000, lower levels of education (less than high school, high school, and some college), being uninsured, and residing in certain census regions (Northeast, Midwest, and South) (Table 1).

Table 2 demonstrates sociodemographic factors associated with telehealth utilization among informal caregivers using a multivariable logistic regression. The results showed that informal caregivers aged 50–64 (OR = 0.36, CI = 0.20–0.65, *p*-value = 0.0011) and those aged 65 and older (OR = 0.40, CI = 0.21–0.74, *p*-value = 0.0048) have significantly lower odds of utilizing telehealth services compared to the reference group of age 35–49. Male informal caregivers had significantly lower odds of utilizing telehealth services compared to their female counterparts (OR = 0.47, CI = 0.25–0.87, *p*-value = 0.0185). When compared to White informal caregivers (the reference group), Black informal caregivers had significantly lower odds of utilizing telehealth services (OR = 0.49, CI = 0.24–0.99, *p*-value = 0.0495). No other racial or ethnic group (Hispanics, Asians, others) showed significant differences compared to Whites. Compared to informal caregivers with incomes of USD 75,000 or more, informal caregivers with household incomes less than USD 20,000 had lower odds of utilizing telehealth services, but this difference did not reach statistical significance (OR = 0.38, CI = 0.14–1.01; *p* = 0.0537). Informal caregivers with health insurance had significantly higher odds of utilizing telehealth services compared to their uninsured counterparts (OR = 5.31, CI = 1.67–16.86, *p*-value = 0.0055). The “census region” variable compared different regions to the West (the reference group). None of the regions (Northeast, Midwest, and South) showed significant differences in telehealth utilization compared to the West.

Table 3 indicates factors associated with perceived good telehealth care among informal caregivers. Compared to informal caregivers with telehealth technical problems, those with no telehealth technical problems had significantly higher odds of perceived good telehealth care (OR = 4.61, CI = 1.61–13.16; *p*-value = 0.0051). Informal caregivers from the South, compared to their counterparts from the west, had significantly higher odds of perceived good telehealth care (OR = 2.95, CI = 1.18–7.37, *p*-value = 0.0213).

## 4. Discussion

The majority of the existing body of knowledge on telehealth utilization and quality, particularly related to sociodemographics, is based on the overall patient population [7,8,9,10,11,12]. Here, among informal caregivers, the comparative analysis between telehealth users and non-users revealed that age, gender, health insurance, and race/ethnicity were significantly associated with telehealth utilization. Among informal caregivers who used telehealth, the perception of good telehealth care was significantly associated with telehealth technical problems and the census region. For the first time, this national study revealed sociodemographic and other characteristics related to telehealth utilization as well as perceived good telehealth among informal caregivers in the USA.

The association of gender, age, health insurance, education, and census regions with telehealth utilization has been frequently reported among overall patients [9,11]. We found that being older than 50 y (vs. 35–50 y), men (vs. women), and uninsured (vs. insured) was significantly associated with lower telehealth utilization among informal caregivers. It was previously revealed that men, compared to women, and patients aged ≥ 65 years, compared to those aged 18–44 years, were significantly less likely to use telehealth [11]. Age can affect one’s comfort with technology and willingness to engage in remote healthcare, while gender may play a role in health-seeking behaviors and communication preferences [11]. Our data on education levels and census regions of informal caregivers showed no statistical significance in terms of telehealth use in multivariable analysis, though descriptive analysis demonstrated a higher percentage of telehealth utilization for college graduates and those living in the West. Prior studies indicated that patients with a bachelor’s degree or above had a 5% higher likelihood of using telehealth than those with only a high school education or less [11]. Another study found that high school graduates used the least telehealth (20.58%), while those with some college (23.29%) or college graduates (22.61%) had similar levels, and those with less than a high school education fluctuated over time [12]. Additionally, telehealth use was highest among those living in the Northeast and West regions [7]. These studies were conducted on large sample sizes of overall patients, enabling the detection of small differences [7,12]. The discrepancy between our results and those of others could be attributed to the current study’s smaller sample size vs. studies with larger sample sizes [7] or to different researched populations (overall patients vs. informal caregivers).

The association of race/ethnicity and income with telehealth utilization is a matter of controversy [7,8,9,10,11,12]. Racial/ethnic disparities can impede access to telehealth services. Income often determines access to the necessary technology and digital literacy, affecting one’s ability to navigate telehealth platforms effectively. A retrospective cohort analysis of ambulatory care in 2020 in Massachusetts revealed that both Black and White patients used telehealth more than Asian patients [8]. A national study indicated that Asian and Hispanic patients were more likely to use telehealth than White and Black patients [11]. Another study indicated that White adult patients were more likely to use telehealth than their Black, Asian, and Hispanic counterparts [7]. Other studies found that East and Southeast Asians used telehealth less than Whites [9], and Black patients had higher levels of use than Asians [12]. We found that Black informal caregivers significantly used less telehealth than their white counterparts. Whether these discrepant results are related to geographic differences or are specific to informal caregivers needs more investigation. Geographical differences in the association of race/ethnicity with other conditions have also been reported in the USA [26]. In terms of income, levels above 200% of the federal poverty line were associated with higher telehealth care utilization among overall patients [7]. However, other studies found that telehealth utilization was highest among low-income households [10,27]. We found no statistical significance in telehealth utilization based on income, though income less than USD 20,000 was nearly significant (Table 2). Our study is the first national finding showing a significant telehealth disparity among Black informal caregivers and an inverse trend among households earning less than USD 20,000 annually.

Telehealth quality/satisfaction has been less studied at the national level [28], and its association with sociodemographic characteristics needs more clarification. A survey of 440 US telehealth patients on satisfaction revealed that sociodemographic characteristics, including female gender, lower education, and income levels (mixed results), were positively associated with telehealth experience satisfaction [13]. A study in Hawaii revealed lower levels of satisfaction during telehealth visits among Asian patients compared to White patients [29]. A survey of 1034 patients in California revealed that satisfaction with telehealth was positively associated with female gender, younger age, non-White ethnicity, and lower education levels, while patient incomes showed mixed trends [16]. Another study found a similar finding with respect to age [18]. A survey of 208 patients in Massachusetts during the pandemic’s first 14 months revealed that non-Whites’ satisfaction was lower than that of Whites [30]. A national study revealed that sociodemographic characteristics, except income, were not associated with telehealth satisfaction among overall patients [28]. Here, among informal caregivers at the national level, we found no significant association between perceived good telehealth care and sociodemographic characteristics, except for census region. The top five telehealth diagnoses and specialists differ between the West and the South [31]. More research is needed to determine whether differences in diagnosis and specialists who visit patients are related to variances in perceived good telehealth care between the South and West.

Telehealth quality/satisfaction has been associated with other characteristics, notably telehealth technological problems, convenience, and avoiding infection exposure [15,18]. In 2020, a study conducted in Southeast Arizona with phone surveys on 562 patients found that overall telehealth satisfaction scores were predicted by distance/convenience and protection against coronavirus exposure [18]. Other studies found that technological problems were inversely associated with telehealth satisfaction [15]. We found that perceived good telehealth care was significantly associated with telehealth technological problems, but not with convenience/distance and avoiding infection exposure among informal caregivers. Our study was conducted in 2022, while other studies were conducted in 2020 in the midst of the pandemic [18], which might have influenced the response to the infection exposure question. Therefore, the differences in results may stem from variations in time periods or study populations, with our focus specifically on informal caregivers as opposed to overall patient groups.

This study’s findings urge engaging a wide range of stakeholders, notably informal caregivers, educational institutions, community organizations, payers, policymakers, technology developers, technology support providers, and researchers. Educational institutions and community organizations should raise telehealth awareness, in particular, among informal caregivers who are male, over 50, and of Black racial/ethnic background. Payers and policymakers should collaborate to make telehealth available to uninsured informal caregivers. Furthermore, technology developers and technology support providers should collaborate to minimize telehealth technical problems, which will eventually improve telehealth quality. Policymakers should also consider these insights to tailor interventions addressing equitable access and enhancing the overall quality of telehealth services for informal caregivers, benefiting their patients as well.

This study has several limitations. The HINTS dataset, while informative, may be subject to selection bias as it relies on voluntary participation, potentially excluding certain demographics. Additionally, the self-reported nature of the data introduces the possibility of recall bias, influencing the accuracy of responses [32]. However, oversampling of certain populations, weighting, and calibration have been incorporated into different steps of the survey building to minimize the impact of these limitations. Moreover, the cross-sectional design of the study limits the establishment of causal relationships, providing associations but not causation between variables. The disparities in telehealth utilization based on race/ethnicity are inconclusive and may vary across different regions and populations [26], warranting further investigation. The analysis of telehealth quality was limited to patient-level variables in the current study, since provider-level data, which could potentially affect telehealth use and its quality, were not available in the HINTS dataset. Any study based on a secondary dataset might be constrained by the predefined variables and metrics present in the secondary dataset, limiting the incorporation of a broader spectrum of assessments [32]. Finally, the limited sample size of informal caregivers in the dataset may have affected the statistical power of the results, highlighting the need for larger, more comprehensive studies in this specific population.

## 5. Conclusions

In conclusion, our findings provide the first national evidence on telehealth use and quality among informal caregivers in the USA after the COVID-19 pandemic. There was a disparity in telehealth use based on age, gender, health insurance, and race/ethnicity. To address these disparities, targeted policies should be defined for older adults, men, the uninsured, and people of the Black race/ethnicity. The perception of good telehealth care was influenced by telehealth technical problems. This is a significant finding, indicating the importance of technology literacy for telehealth quality at the national level. Addressing these issues is crucial for enhancing telehealth services for this essential segment of the healthcare workforce, benefiting patients as well, and improving disease outcomes.

## Figures and Tables

**Table 1 healthcare-11-03193-t001:** Characteristics of informal caregivers by telehealth utilization in USA (2022 HINTS *).

Characteristics (%)	Telehealth Users (N = 421) **	Non-Telehealth Users (N = 397) ***
**Age groups, years**		
18–34	10.4	8.7
35–49	24.4	19.9
50–64	34.0	37.6
65+	31.0	33.6
**Gender**		
Male	30.3	33.3
Female	69.6	66.6
**Race/Ethnicity**		
White	52.6	54.7
Black	15.1	19.9
Hispanic	21.7	13.3
Asian	5.8	6.1
Others	4.5	5.8
**Household income levels**		
<USD 20,000	13.5	17.1
USD 20,000–less than USD 35,000	13.0	10.9
USD 35,000–less than USD 50,000	14.1	13.6
USD 50,000–less than USD 75,000	17.9	20.1
≥USD 75,000	41.2	38.1
**Education**		
Less than high school	5.0	5.81
High school	14.5	15.9
Some college	29.7	32.8
College Graduate or More	50.6	45.4
**Insured**		
Yes	94.6	88.7
No	5.3	11.2
**Census Regions**		
Northeast	14.7	17.0
Midwest	14.2	18.5
South	42.4	46.3
West	28.5	18.0

* HINTS, Health Information and National Trends Survey. ** Frequency of missing values in all groups was less than 5%, except race/ethnicity (39 missing values) and income levels (44 missing values). *** Frequency of missing values in all groups was less than 5%, except race/ethnicity (34 missing values) and income levels (43 missing values).

**Table 2 healthcare-11-03193-t002:** Sociodemographic factors associated with telehealth utilization among informal caregivers in the USA (2022 HINTS *).

Predictors	Odds Ratio	CI **	*p*-Value
**Age groups, years**			
Reference, 35–49			
18–34	0.43	0.13–1.42	0.1659
50–64	0.36	0.20–0.65	0.0011
65+	0.40	0.21–0.74	0.0048
**Gender**			
Female (reference)			
Male	0.47	0.25–0.87	0.0185
**Race/Ethnicity**			
White (reference)			
Black	0.49	0.24–0.99	0.0495
Hispanic	1.41	0.62–3.19	0.4008
Asians	0.50	0.10–2.53	0.4004
Others	1.50	0.62–3.65	0.3579
**Household income levels**			
≥USD 75,000 (reference)			
<USD 20,000	0.38	0.14–1.01	0.0537
USD 20,000–less than USD 35,000	1.92	0.73–5.03	0.1786
USD 35,000–less than USD 50,000	0.84	0.73–1.87	0.6718
USD 50,000–less than USD 75,000	0.72	0.26–1.97	0.5207
**Education**			
≥College graduate (reference)			
Less than high school	2.94	0.52–16.43	0.2123
High school	1.29	0.48–3.47	0.5948
Some college	1.10	0.62–1.93	0.7264
**Health insurance**			
No (reference)			
Yes	5.31	1.67–16.86	0.0055
**Census region**			
West (reference)			
Northeast	1.68	0.73–3.88	0.2129
Midwest	0.95	0.33–2.68	0.9272
South	1.45	0.72–2.95	0.2870

* HINTS, Health Information National Trends Survey. ** CI, confidence interval.

**Table 3 healthcare-11-03193-t003:** Factors associated with perceived good telehealth care among informal caregivers in the USA (2022 HINTS *).

Predictors	Odds Ratio	CI **	*p*-Value
**Age groups, years**			
Reference, 35–49			
18–34	0.78	0.09–6.69	0.8232
50–64	0.62	0.20–1.90	0.3986
65+	0.324	0.06–1.70	0.1784
**Gender**			
Female (reference)			
Male	0.57	0.15–2.10	0.3984
**Race/Ethnicity**			
White (reference)			
Black	0.84	0.19–3.77	0.8213
Hispanic	1.24	0.40–3.88	0.6961
Asians	0.4	0.08–1.96	0.2557
Others	0.53	0.05–5.84	0.5978
**Household income levels**			
≥USD 75,000 (reference)			
<Less than USD 20,000	1.60	0.29–8.71	0.5781
USD 20,000–less than USD 35,000	0.76	0.08–6.62	0.8030
USD 35,000–less than USD 50,000	2.72	0.58–12.78	0.1988
USD 50,000–less than USD 75,000	1.24	0.33–4.60	0.7354
**Education**			
≥College graduate (reference)			
Less than high school	0.19	0.01–2.68	0.2186
High school	1.86	0.46–7.49	0.3732
Some college	1.47	0.63–3.42	0.3635
**Health insurance**			
No (reference)			
Yes	1.67	0.23–12.15	0.6027
**Census region**			
West (reference)			
Northeast	0.96	0.29–3.19	0.9496
Midwest	1.92	0.44–8.40	0.3777
South	2.95	1.18–7.37	0.0213
**Problem with telehealth**			
Problems with telehealth (reference)			
No telehealth problem	4.61	1.61–13.16	0.0051
**Telehealth Convenience**			
No (reference)			
Yes	1.19	0.43–3.27	0.7253
**Concerns with infection exposure**			
No (reference)			
Yes	1.60	0.72–3.54	0.2362

* HINTS, Health Information National Trends Survey. ** CI, confidence interval.

## Data Availability

The data is publicly available via link https://hints.cancer.gov/data/survey-instruments.aspx (accessed on 16 November 2023).

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
