# Peer review of "Telehealth Utilization and Good Care among Informal Caregivers: Health Information National Trends Survey, 2022"

_healthcare, 2023, doi:10.3390/healthcare11243193_

Round 1

Reviewer 1 Report

Comments and Suggestions for Authors

First of all I would like to thank the authors for their work and the editor for allowing me to review this article. Below I highlight some positive aspects or aspects that could be improved. 

Positive Aspects:

Topic Relevance: The choice of topic is relevant, especially in the current context of the COVID-19 pandemic, which has increased the promotion of telemedicine.

Methodological Approach: The use of HINTS data and the statistical analysis approach provides a sound basis for the research. In addition, the inclusion of sociodemographic and quality of service variables allows for a comprehensive analysis.

Areas for Improvement:

Clarity in the Introduction: The article's introduction is lengthy and could benefit from greater clarity and conciseness. It would be useful to reduce the amount of information and focus on the objectives and importance of the study.

Data Contextualization: The explanation of the HINTS data selection and methodology could be more detailed to ensure the reader's understanding of the validity and representativeness of the results.

Justification of Variables: The justification for the selection of certain variables could be more robust. For example, it is mentioned that quality of service variables (technical problems, convenience, exposure to infections) were included, but the rationale behind these choices could be addressed in more detail.

Discussion of Limitations: Although some limitations are mentioned, the discussion could be expanded to address possible biases and methodological constraints that could affect the interpretation of the results.

Writing and Formatting: There are some grammatical and editorial errors that could be corrected to improve the flow of the article.

Overall, the study provides valuable insight into telemedicine among informal caregivers, but could benefit from adjustments in presentation and clarity to improve its impact and accessibility.

Author Response

Dear Editor,

Dear reviewers,

Thank you for reviewing the manuscript and sending your valuable comments. Please find below point-by-point responses to these comments. Changes have been highlighted in the text.

Reviewer 1:

First of all I would like to thank the authors for their work and the editor for allowing me to review this article. Below I highlight some positive aspects or aspects that could be improved. 

 Positive Aspects:

 Topic Relevance: The choice of topic is relevant, especially in the current context of the COVID-19 pandemic, which has increased the promotion of telemedicine.

 Methodological Approach: The use of HINTS data and the statistical analysis approach provides a sound basis for the research. In addition, the inclusion of sociodemographic and quality of service variables allows for a comprehensive analysis.

Areas for Improvement:

 Clarity in the Introduction: The article's introduction is lengthy and could benefit from greater clarity and conciseness. It would be useful to reduce the amount of information and focus on the objectives and importance of the study.

Reply: Thank you for the comment. The introduction section was improved. More than 10 sentences were deleted or combined in the introduction section.

Data Contextualization: The explanation of the HINTS data selection and methodology could be more detailed to ensure the reader's understanding of the validity and representativeness of the results.

Reply: Thank you for the comment. The Survey section under Methods was revised as below:

The HINTS 6 employed a rigorous methodology for sample selection, data collection, data management, and weighting and variance estimation [23, 24].

The HINTS 6 survey employed a two-stage sampling strategy for sample selection. A stratified sample of addresses was initially chosen from a residential address file. In the subsequent stage, one adult was selected within each sampled household. Unlike previous HINTS samples, HINTS 6 featured an expanded sample design that included four sampling strata by further categorizing the traditional high and low minority strata based on rural and urban geographic areas. While the high minority strata were still sampled at a higher rate, the rural strata among the low minority strata were also sampled at a higher rate than the urban strata, ensuring an adequate representation of rural addresses in the overall sample [23, 24].

Selected individuals were categorized into control and treatment groups. In the control group, respondents were given the choice to respond via paper or online, whereas the treatment group initially had online-only options, later incorporating paper surveys in subsequent mailings. The survey was available in both English and Spanish, and all participants, irrespective of the response mode, received a $2 pre-paid monetary incentive to boost participation. Additional incentives were provided to the control group for completing the survey online. Mail surveys offer a cost-effective means to reach a diverse population, provide flexibility for respondents to complete the survey at their convenience, and accommodating those without telephone access or those who prefer a private setting, leading to enhanced response rates and a representative study sample. The choice of online surveys was added in 2019 to examine the feasibility of encouraging a sufficient number of participants to transition to online data collection while sustaining, or even enhancing, response rates. The mailing procedures are detailed in the HINTS 6 methodological report [23, 24].

For data management, each retuned paper questionnaire underwent scanning, verification, cleaning, and editing procedures. Simultaneously, each web questionnaire underwent assessment using the Survey Builder verification and cleaning processes, and the data were extracted into a unified database, subsequently edited alongside the paper data. Additionally, imputation procedures were applied to both web and paper data [24]. The final HINTS 6 sample consists of 6,252 respondents; of them; 67 were considered partial completers who did not answer the entire survey (responded to 50%–79% of questions) [24]. A questionnaire was considered to be complete if at least 80% of questions were answered. The overall household response rate was 28.1%. Further details on sampling are publicly available in the HINTS 6 Methodology Report [24].

Each surveyed adult was assigned a full-sample weight along with a set of 50 replicate weights. Replicate weights were generated using the 'delete one' jackknife (JK1) replication method [24]. The full-sample weight and replicate weights were utilized for estimating population/subpopulation estimates and standard errors for these estimates, respectively. The application of sampling weights ensured accurate inferences from the respondent sample to the broader population, addressing nonresponse and noncoverage biases as much as possible. Calibration of person-level weights to population counts (referred to as control totals) was performed using estimates from the United States Census Bureau's 2021 American Community Survey with the following variables: age, gender, education, marital status, race, ethnicity, and census region [24]. The 2021 National Health Interview Survey was also used to calibrate HINTS 6 data control totals regarding percent with health insurance [24].

Justification of Variables: The justification for the selection of certain variables could be more robust. For example, it is mentioned that quality of service variables (technical problems, convenience, exposure to infections) were included, but the rationale behind these choices could be addressed in more detail.

 Reply: Thank you for the comment. The following sentences were added to the Introduction section and Limitation section:

To the introduction:

Patient satisfaction with telehealth has also been linked to comfort with technology, the convenience it offers, and the ability to avoid potential infection exposure [16, 17]. Telehealth technical problems could disrupt the smooth flow of communication between parties, resulting in delayed or interrupted virtual sessions, affecting the quality of telehealth and patient satisfaction [16, 17]. The convenience and reduced travel distances can contribute to the overall quality of healthcare delivery by promoting more frequent and timely interactions between caregivers and healthcare providers [16, 17]. The avoidance of crowded waiting rooms and unnecessary exposure to potential sources of infection can also positively influence telehealth patient satisfaction [18].

To the limitations:

The analysis of telehealth quality was limited to patient-level variables in the current study since provider-level data, which could potentially affect telehealth use and its quality, were not available in the HINTS dataset. Any study based on a secondary dataset might be constrained by the predefined variables and metrics present in the secondary dataset, limiting the incorporation of a broader spectrum of assessments [32].

Discussion of Limitations: Although some limitations are mentioned, the discussion could be expanded to address possible biases and methodological constraints that could affect the interpretation of the results.

 Reply: Thank you for the comment. The following sentences were added to the limitation section:

The HINTS dataset, while informative, may be subject to selection bias as it relies on voluntary participation, potentially excluding certain demographics. Additionally, the self-reported nature of the data introduces the possibility of recall bias, influencing the accuracy of responses [32]. However, oversampling of certain populations, weighting, and calibration have been incorporated into different steps of the survey building to minimize the impact of these limitations. Moreover, the cross-sectional design of the study limits the establishment of causal relationships, providing associations but not causation between variables.

Writing and Formatting: There are some grammatical and editorial errors that could be corrected to improve the flow of the article.

Reply: Thank you for the comment. Grammatical errors were corrected.

 Overall, the study provides valuable insight into telemedicine among informal caregivers, but could benefit from adjustments in presentation and clarity to improve its impact and accessibility.

Reviewer 2 Report

Comments and Suggestions for Authors

In line 206, it is stated that 831 informal caregivers were included in the study. However, the sum of the value of n in the two columns of the table is 818 (421+397).

Although the article is informative for the reader regarding the sociodemographic variables that are associated with the use of telehealth services, why a multivariate model was not tested? Perhaps this could be addressed in the discussion/limitations.

Author Response

Dear Editor,

Dear reviewers,

Thank you for reviewing the manuscript and sending your valuable comments. Please find below point-by-point responses to these comments. Changes have been highlighted in the text.

Reviewer 2

In line 206, it is stated that 831 informal caregivers were included in the study. However, the sum of the value of n in the two columns of the table is 818 (421+397).

Reply: Thank you for the comment. The following sentences were added/revised in Result section and Method section:

Result section:

Totally, 831 informal caregivers were available in the dataset. Of them, 13 had missing values for telehealth.

Method section:

Missing values were replaced with dots (.) using SAS codes [23].

Although the article is informative for the reader regarding the sociodemographic variables that are associated with the use of telehealth services, why a multivariate model was not tested? Perhaps this could be addressed in the discussion/limitations.

Reply: Thank you for the comment. We believe Table 2 might have addressed this comment. We highlighted it in the result section, as below:

Table 2 demonstrates sociodemographic factors associated with telehealth utilization among informal caregivers using a multivariable logistic regression.

Reviewer 3 Report

Comments and Suggestions for Authors

Author Response

Dear Editor,

Dear reviewers,

Thank you for reviewing the manuscript and sending your valuable comments. Please find below point-by-point responses to these comments. Changes have been highlighted in the text.

Reviewer 3

The manuscript addresses a very current, pertinent, and significant topic for healthcare. Interesting title that reflects the object of study. The summary is appropriate in terms of information regarding the study developed. The problem is presented in a consistent manner, with the presentation of studies that express the state of the art in relation to the quality of telehealth services and satisfaction with telehealth. It presents the objectives of the study and highlights the importance of this study for the use and quality of health services. In the methodology, it presents the type of study and sampling, in addition to describing the variables used. It states that the questions used in the study were recently added to the National Survey, namely the Survey on National Health Information Trends (HINTS).

It should be evident to readers (from all over the world) that this survey and the different components of it offer guarantees of credibility, that is, information regarding the reliability and validity of the information production instrument. And this is not evident in the methodological description.

Reply: Thank you for the comment. The section of Survey under method was revised as below:

Launched in 2003, HINTS is a nationally recognized survey conducted periodically by the National Cancer Institute (NCI). Its target population is civilian, non-institutionalized adults aged ≥18 residing in the US [24]. The HINTS 6 (March 7, 2022, to November 8, 2022) contains 475 questions in 17 sections, including section A (looking for health information), section B (using the internet to find information), section C (your health care), section D (telehealth), section E (medical records), section F (caregiving), section G (genetic testing), section H (your overall health), section J (environment and health), section K (social determinants of health), section L (health and nutrition), section M (physical activity and exercise), section N (tobacco products), sections O, P, and Q (cancer-related questions), and section R (you and your household) [24]. The HINTS 6 employed a rigorous methodology for sample selection, data collection, data management, and weighting and variance estimation [23, 24].

The HINTS 6 survey employed a two-stage sampling strategy for sample selection. A stratified sample of addresses was initially chosen from a residential address file. In the subsequent stage, one adult was selected within each sampled household. Unlike previous HINTS samples, HINTS 6 featured an expanded sample design that included four sampling strata by further categorizing the traditional high and low minority strata based on rural and urban geographic areas. While the high minority strata were still sampled at a higher rate, the rural strata among the low minority strata were also sampled at a higher rate than the urban strata, ensuring an adequate representation of rural addresses in the overall sample [23, 24].

Selected individuals were categorized into control and treatment groups. In the control group, respondents were given the choice to respond via paper or online, whereas the treatment group initially had online-only options, later incorporating paper surveys in subsequent mailings. The survey was available in both English and Spanish, and all participants, irrespective of the response mode, received a $2 pre-paid monetary incentive to boost participation. Additional incentives were provided to the control group for completing the survey online. Mail surveys offer a cost-effective means to reach a diverse population, provide flexibility for respondents to complete the survey at their convenience, and accommodating those without telephone access or those who prefer a private setting, leading to enhanced response rates and a representative study sample. The choice of online surveys was added in 2019 to examine the feasibility of encouraging a sufficient number of participants to transition to online data collection while sustaining, or even enhancing, response rates. The mailing procedures are detailed in the HINTS 6 methodological report [23, 24].

For data management, each retuned paper questionnaire underwent scanning, verification, cleaning, and editing procedures. Simultaneously, each web questionnaire underwent assessment using the Survey Builder verification and cleaning processes, and the data were extracted into a unified database, subsequently edited alongside the paper data. Additionally, imputation procedures were applied to both web and paper data [24]. The final HINTS 6 sample consists of 6,252 respondents; of them; 67 were considered partial completers who did not answer the entire survey (responded to 50%–79% of questions) [24]. A questionnaire was considered to be complete if at least 80% of questions were answered. The overall household response rate was 28.1%. Further details on sampling are publicly available in the HINTS 6 Methodology Report [24].

Each surveyed adult was assigned a full-sample weight along with a set of 50 replicate weights. Replicate weights were generated using the 'delete one' jackknife (JK1) replication method [24]. The full-sample weight and replicate weights were utilized for estimating population/subpopulation estimates and standard errors for these estimates, respectively. The application of sampling weights ensured accurate inferences from the respondent sample to the broader population, addressing nonresponse and noncoverage biases as much as possible. Calibration of person-level weights to population counts (referred to as control totals) was performed using estimates from the United States Census Bureau's 2021 American Community Survey with the following variables: age, gender, education, marital status, race, ethnicity, and census region [24]. The 2021 National Health Interview Survey was also used to calibrate HINTS 6 data control totals regarding percent with health insurance [24].

 On the other hand, I did not find information regarding ethical aspects (authorizations/ethics declarations).

Reply: Thank you for the comment. The sentence was highlighted in the method section as below:

The University of Nevada Las Vegas deemed this secondary data analysis of deidentified, publicly available data exempt from review.

Being a cross-sectional study, the existence of cause-and-effect relationships between the variables under study is not expected. The language of dependent and independent variables is typical of experimental studies. However, the authors emphasized the use of independent variables in statistical analysis. It seems to us that the nomenclature of independent variables is clearly unnecessary.

Reply: Thank you for the comment. Independent variables were replaced by predictors, and dependent variables were replaced by “outcome”. Moreover, following sentences were added to the limitations of the study:

Moreover, the cross-sectional design of the study limits the establishment of causal relationships, providing associations but not causation between variables.

In the study in question, these are variables that characterize the subjects under study and are associated with the key variables of the study. The results are well presented, both in descriptive terms and with the use of tables (they show the data correctly and in an appropriate way to facilitate analysis by the reader). They are well supported statistically. The discussion is interesting and allows for a very consistent interpretation of the study findings, valuing the association relationships that were tested in this study. The implications (political and educational) that arise from carrying out the study are highlighted. Despite the synthesis that emerges at the end of the discussion, a conclusion that adequately responds to the objectives defined for the study in question seems pertinent. The references are all from the last five years. In short, the study has quality, but conclusion needs to be improved methodologically and with a conclusion that is adequate (consistent with the study findings) and interesting for the magazine's readers

Reply: Thank you for the comment. The conclusion section was revised as below:

In conclusion, our findings provide the first national evidence on telehealth use and quality among informal caregivers in the USA after the COVID-19 pandemic. There was a disparity in telehealth use based on age, gender, health insurance, and race/ethnicity. To address these disparities, targeted policies should be defined for older adults, men, the uninsured, and people of the black race/ethnicity. The perception of good telehealth care was influenced by telehealth technical problems. This is a significant finding, indicating the importance of technology literacy for telehealth quality at the national level. Addressing these issues is crucial for enhancing telehealth services for this essential segment of the healthcare workforce, benefiting patients as well, and improving disease outcomes.

Reviewer 4 Report

Comments and Suggestions for Authors

This manuscript is interesting and timely issue given the ever increasing Telehealth. However, there are several issues that limit the contribution of the manuscript and need to be addressed by the authors.

It would be useful if the authors provided some more background on the HINTS survey for readers not familiar with this database. In addition, it would be interesting to note why mail was chosen to gather the survey information and not combined with other techniques such as telephone surveys and online surveys.

Methods:

Please add some informations on the general health of the sample or health conditions.

I think it would be beneficial to the reader to include the survey as an Additional file. Please insert the link to the Additional file in the ‘measures’ section.

Insert how many questions/ number of sections were in the survey and give more detail regarding what the respondent completed in the measures section.

Analysis:

Please insert information on which regression model was used. This information is currently incorrectly placed in the ‘results’ section.

How was the missing data dealt with? Please comment in the methods section.

Results:

Please separate the ‘Results and Discussion’ section in to separate sections to adhere to the journal’s standards for reporting research data. The ‘Discussion’ section should start when the author’s begin to interpret the data. 

Discussion

I suggest the authors discuss the importance of stakeholder engagement and link it to the appropriateness of health questions or the implementation of recommendations.

Conclusion:

 I think the conclusion is currently quite weak. It should be more interpretive.

Keywords: 

Add more keywords to reflect the design of the research study and the population to increase the visibility of the article for those with similar research interests.

Author Response

Dear Editor,

Dear reviewers,

Thank you for reviewing the manuscript and sending your valuable comments. Please find below point-by-point responses to these comments. Changes have been highlighted in the text.

Reviewer 4

This manuscript is interesting and timely issue given the ever-increasing Telehealth. However, there are several issues that limit the contribution of the manuscript and need to be addressed by the authors.

It would be useful if the authors provided some more background on the HINTS survey for readers not familiar with this database. In addition, it would be interesting to note why mail was chosen to gather the survey information and not combined with other techniques such as telephone surveys and online surveys.

Reply: Thank you for the comment. The method section was improved as below to give more background on HINTS and data collection methods (data collection is highlighted in gray):

The HINTS 6 employed a rigorous methodology for sample selection, data collection, data management, and weighting and variance estimation [23, 24].

The HINTS 6 survey employed a two-stage sampling strategy for sample selection. A stratified sample of addresses was initially chosen from a residential address file. In the subsequent stage, one adult was selected within each sampled household. Unlike previous HINTS samples, HINTS 6 featured an expanded sample design that included four sampling strata by further categorizing the traditional high and low minority strata based on rural and urban geographic areas. While the high minority strata were still sampled at a higher rate, the rural strata among the low minority strata were also sampled at a higher rate than the urban strata, ensuring an adequate representation of rural addresses in the overall sample [23, 24].

Selected individuals were categorized into control and treatment groups. In the control group, respondents were given the choice to respond via paper or online, whereas the treatment group initially had online-only options, later incorporating paper surveys in subsequent mailings. The survey was available in both English and Spanish, and all participants, irrespective of the response mode, received a $2 pre-paid monetary incentive to boost participation. Additional incentives were provided to the control group for completing the survey online. Mail surveys offer a cost-effective means to reach a diverse population, provide flexibility for respondents to complete the survey at their convenience, and accommodating those without telephone access or those who prefer a private setting, leading to enhanced response rates and a representative study sample. The choice of online surveys was added in 2019 to examine the feasibility of encouraging a sufficient number of participants to transition to online data collection while sustaining, or even enhancing, response rates. The mailing procedures are detailed in the HINTS 6 methodological report [23, 24].

For data management, each retuned paper questionnaire underwent scanning, verification, cleaning, and editing procedures. Simultaneously, each web questionnaire underwent assessment using the Survey Builder verification and cleaning processes, and the data were extracted into a unified database, subsequently edited alongside the paper data. Additionally, imputation procedures were applied to both web and paper data [24]. The final HINTS 6 sample consists of 6,252 respondents; of them; 67 were considered partial completers who did not answer the entire survey (responded to 50%–79% of questions) [24]. A questionnaire was considered to be complete if at least 80% of questions were answered. The overall household response rate was 28.1%. Further details on sampling are publicly available in the HINTS 6 Methodology Report [24].

Each surveyed adult was assigned a full-sample weight along with a set of 50 replicate weights. Replicate weights were generated using the 'delete one' jackknife (JK1) replication method [24]. The full-sample weight and replicate weights were utilized for estimating population/subpopulation estimates and standard errors for these estimates, respectively. The application of sampling weights ensured accurate inferences from the respondent sample to the broader population, addressing nonresponse and noncoverage biases as much as possible. Calibration of person-level weights to population counts (referred to as control totals) was performed using estimates from the United States Census Bureau's 2021 American Community Survey with the following variables: age, gender, education, marital status, race, ethnicity, and census region [24]. The 2021 National Health Interview Survey was also used to calibrate HINTS 6 data control totals regarding percent with health insurance [24].

Please add some information on the general health of the sample or health conditions.

Reply: We appreciate your comment. While our primary research question did not include general health, we recognize that caregivers may have interests in engaging with telehealth for both their personal well-being and that of their patients. Investigating this aspect may be considered in future studies, possibly with a larger sample size and more diverse subgroups. Thank you for bringing this perspective to our attention.

I think it would be beneficial to the reader to include the survey as an Additional file. Please insert the link to the Additional file in the ‘measures’ section.

Reply: Thank you for the comment. Two references were included (23, 24). One reference (23) is a link to the HINTS website, which contains all survey instruments from 2003 to present.

The other reference is a direct link to the HINTS 6 methodology report (24).

  1. National Cancer Institute. The Health Information National Trends Survey (Hints) https://hints.cancer.gov/data/survey-instruments.aspx. 2023.
  2. Westat Health Information National Trends Survey 6 (Hints 6). Https://Hints.Cancer.Gov/Docs/Methodologyreports/Hints_6_Methodologyreport.Pdf; 2023.

Insert how many questions/ number of sections were in the survey and give more detail regarding what the respondent completed in the measures section.

Reply: Thank you for the comment. They were added to the Method section as below:

The first paragraph of method section under Survey:

The HINTS 6 (March 7, 2022, to November 8, 2022) contains 475 questions in 17 sections, including section A (looking for health information), section B (using the internet to find information), section C (your health care), section D (telehealth), section E (medical records), section F (caregiving), section G (genetic testing), section H (your overall health), section J (environment and health), section K (social determinants of health), section L (health and nutrition), section M (physical activity and exercise), section N (tobacco products), sections O, P, and Q (cancer-related questions), and section R (you and your household) [24]. The HINTS 6 employed a rigorous methodology for sample selection, data collection, data management, and weighting and variance estimation [23, 24].

To the fourth paragraph of Method section:

The final HINTS 6 sample consists of 6,252 respondents; of them; 67 were considered partial completers who did not answer the entire survey (responded to 50%–79% of questions) [24]. A questionnaire was considered to be complete if at least 80% of questions were answered. The overall household response rate was 28.1%. Further details on sampling are publicly available in the HINTS 6 Methodology Report [24].

Analysis:

Please insert information on which regression model was used. This information is currently incorrectly placed in the ‘results’ section.

Reply: Thank you for the comment. The following sentence was highlighted in the method section:

For both outcomes, weighted multivariable logistic regression models (proc surveylogistic) were used to calculate odds ratios (ORs) and 95% confidence intervals (CIs).

How was the missing data dealt with? Please comment in the methods section.

Reply: Thank you for the comment. It was added to the method section:

Missing values were replaced with dots (.) using SAS codes [23].

Results:

Please separate the ‘Results and Discussion’ section in to separate sections to adhere to the journal’s standards for reporting research data. The ‘Discussion’ section should start when the author’s begin to interpret the data. 

Reply: Thank you for the comment. It was done.

Discussion

I suggest the authors discuss the importance of stakeholder engagement and link it to the appropriateness of health questions or the implementation of recommendations.

Reply: Thank you for the comment. This paragraph was revised in the discussion section:

This study's findings urge engaging a wide range of stakeholders, notably informal caregivers, educational institutions, community organizations, payers, policymakers, technology developers, technology support providers, and researchers. Educational institutions and community organizations should raise telehealth awareness, in particular, among informal caregivers who are male, over 50, and of black racial/ethnic background. Payers and policymakers should collaborate to make telehealth available to uninsured informal caregivers. Furthermore, technology developers and technology support providers collaborate to minimize telehealth technical problems, which will eventually improve telehealth quality. Policymakers should also consider these insights to tailor interventions addressing equitable access and enhancing the overall quality of telehealth services for informal caregivers, benefiting their patients as well.

Conclusion:

 I think the conclusion is currently quite weak. It should be more interpretive.

Reply: Thank you for the comment. The conclusion was revised as below:

In conclusion, our findings provide the first national evidence on telehealth use and quality among informal caregivers in the USA after the COVID-19 pandemic. There was a disparity in telehealth use based on age, gender, health insurance, and race/ethnicity. To address these disparities, targeted policies should be defined for older adults, men, the uninsured, and people of the black race/ethnicity. The perception of good telehealth care was influenced by telehealth technical problems. This is a significant finding, indicating the importance of technology literacy for telehealth quality at the national level. Addressing these issues is crucial for enhancing telehealth services for this essential segment of the healthcare workforce, benefiting patients as well, and improving disease outcomes.

Keywords: 

Add more keywords to reflect the design of the research study and the population to increase the visibility of the article for those with similar research interests.

Reply: Thank you for the comment. More keywords were added:

Keywords: Disparity, Informal caregivers, HINTS, Quality, Race, Surveys, Telehealth.